# Soliton Microcomb on Chip Integrated *Si*_3_N_4_ Microresonators with Power Amplification in Erbium-Doped Optical Mono-Core Fiber

**DOI:** 10.3390/mi13122125

**Published:** 2022-11-30

**Authors:** Xinpeng Chen, Suwan Sun, Weizhu Ji, Xingxing Ding, You Gao, Tuo Liu, Jianxiang Wen, Hairun Guo, Tingyun Wang

**Affiliations:** Key Laboratory of Specialty Fiber Optics and Optical Access Networks, Joint International Research Laboratory of Specialty Fiber Optics and Advanced Communication, Shanghai University, Shanghai 200444, China

**Keywords:** soliton microcombs, dispersion, optical amplification

## Abstract

Soliton microcombs, offering large mode spacing and broad bandwidth, have enabled a variety of advanced applications, particularly for telecommunications, photonic data center, and optical computation. Yet, the absolute power of microcombs remains insufficient, such that optical power amplification is always required. Here, we demonstrate a combined technique to access power-sufficient optical microcombs, with a photonic-integrated soliton microcomb and home-developed erbium-doped gain fiber. The soliton microcomb is generated in an integrated ${\mathrm{Si}}_{3}N_{4}$ microresonator chip, which serves as a full-wave probing signal for power amplification. After the amplification, more than 40 comb modes, with 115-GHz spacing, reach the onset power level of >−10 dBm, which is readily available for parallel telecommunications , among other applications.

## 1. Introduction

Soliton microcombs in miniatured and chip-scale optical microresonators have unprecedentedly widened the performance of optical frequency combs and have enabled advanced applications (massive parallel telecommunication [1,2,3,4,5], optical ranging [6,7], massive parallel LIDAR [8], ultrafast photonic data switching [9,10], and photonic neuromorphic computation [11,12], among others [13,14,15,16,17]). In the context of telecommunication, the microwave repetition rate underlying the soliton microcombs, typically 10–1000 GHz, has made them naturally suitable to serve as the multi-wavelength laser source that is desired to implement high-volume and high-speed communication systems by means of dense wavelength division multiplexing (DWDM, cf. ITU Standard G.694.1 (10/20)). Compared with the conventional multiple-laser array, or electro-optically modulated lasers (also called EO comb), soliton microcombs demonstrated shown unbeatable advantages, including having a large bandwidth providing a large number of optical channels for parallel communication [18], and the capability to be chip-integrated and therefore mass produced with high compactness.

Great advances in soliton microcombs have been witnessed in the last decade [19,20]. While a number of photonic platforms supporting the generation of soliton microcombs [18,21,22,23,24,25,26,27] have emerged, silicon nitride (${\mathrm{Si}}_{3}N_{4}$) has been recognized as one of the most successful materials that features both decent optical properties and CMOS compatibility [28]. This allows for the wafer-scale fabrication of soliton microcomb photonic chips on ${\mathrm{Si}}_{3}N_{4}$ [29], and the process is currently foundry-available. In detail, ${\mathrm{Si}}_{3}N_{4}$ has wide transparency from visible to the mid-infrared, free from two-photon absorption, and has a high purity of spontaneous nonlinear effects (i.e., Kerr nonlinearity with weak Raman effects). The nonlinear efficiency is almost ten times higher than the silica. Successful fabrication receipts [30,31,32,33,34,35] have been developed for high-quality and high-transmission ${\mathrm{Si}}_{3}N_{4}$ photonic waveguides, enabling chip-scale microresonators that are of an ultra-high quality (*Q*) factor.

The generation of soliton microcombs in such high-*Q* and nonlinear microresonators is usually non-trivial as an essential energy building-up process is required to support the nonlinear four wave mixing as well as the formation of dissipative temporal solitons as local structures in the cavity. The soliton state is fundamentally in the regime of the double balance between the cavity dispersion and the nonlinearity, and between the cavity dissipation and the gain seeded by an external pumping source [19,36]. It is accessed in the red-detuned condition of the pumping frequency with respect to the cavity resonance [37,38,39]. Therefore, an effective laser tuning process, from the cavity resonance to the red-detuned side, is required. In this regard, a variety of schemes have been developed to the access of soliton microcombs, including the classic laser frequency tuning method [39], EO modulated sideband tuning [40], or equivalently the cavity resonance tuning via thermal or piezo-electric effect [41,42]. Notably, laser self-injection locking with the high-*Q* microresonator is another effective scheme [43,44,45,46,47], which fundamentally allows for the turn-key operation of soliton microcombs [45] and technically opens the access to monolithic microcomb chips with hybrid integrated laser diodes [48,49,50]. Yet, the frequency tuning scenario remains unchanged, which is underlying the laser switch-on with spontaneous frequency and power pulling. Moreover, the soliton access range is also cut by the cavity thermal effect, given that a power transition is usually featured from the noise state in the regime of the modulation instability (high power) to the stabilized soliton state (relatively low power) during the laser tuning process. As such, several solutions are found to counterbalance the overall power change in the cavity [51,52,53,54,55,56], which are helpful to recover the soliton access.

To date, soliton microcombs have served not only in applications but also in the study of dissipative soliton physics. A number of soliton dynamics have been discovered and investigated, including soliton-induced dispersive wave generation [18,55,57,58], the soliton double-resonance feature [59], the soliton breathing dynamic [60,61,62,63], the soliton-induced Raman frequency shift [64,65], and the soliton-induced microwave purification [16,66]. Moreover, there is a trend to the study of the soliton dynamics in complex or coupled resonator systems beyond the mono-cavity structure [67,68,69,70,71,72,73]. In this context, the most exciting result regards the boosted power efficiency of soliton microcombs [67,72,73], particularly the bright soliton in the anomalous dispersion regime where a record high efficiency approaching 55% has been reported [72].

Yet, the absolute power of soliton microcombs remains insufficient for most of the applications, and power amplification is therefore needed. Conventional schemes are mainly based on rare-earth-doped fibers that would introduce a remarkable power gain (typ. >25 dB) in certain wavebands. Recently, photonic-integrated optical amplifiers have been demonstrated, by either means of optical parametric amplification [74] or active doping [75]. While such schemes have potentially paved the way for fully integrated power-amplified microcombs, the current gain or bandwidth is not as good as the gain-fiber solution, and further improvements are necessarily required to reduce the loss rate of photonic waveguides and increase the doping rate of active elements into the waveguide.

In this paper, we showcase the soliton microcomb generation on chip integrated ${\mathrm{Si}}_{3}N_{4}$ microresonators that are foundry-available, and we investigate the full wave amplification of the microcomb with a home-developed erbium-doped gain fiber, in which more than 40 comb modes (ca. 35nm in the wavelength span) are power-amplified to above the level of −10 dBm, readily for WDM.

## 2. Results

### 2.1. Characterization of Photonic Chip Integrated ${\mathrm{Si}}_{3}N_{4}$ Microresonators

In our experiment, ${\mathrm{Si}}_{3}N_{4}$ microresonator chips (Figure 1a) are fabricated at the foundry LIGENTECH, in which each resonator consists of a ring waveguide as the cavity and a bus waveguide for evanescent wave coupling to the cavity. The waveguide core is ${\mathrm{Si}}_{3}N_{4}$, and the cladding is ${\mathrm{SiO}}_{2}$.

While the waveguide core height is usually fixed, the width as well as the ring radius can be lithographically tuned. In a selected microresonator, the ring waveguide has a radius of 220μm and has a close-to-rectangular cross section that is 1800 × 780 nm${}^{2}$. This resonator is first characterized in terms of the dispersion measurement and the resonance linewidth measurement, by means of calibrated tunable laser spectroscopy [76,77]. The experimental setup is shown in Figure 1b, in which a narrow-linewidth tunable laser (Toptica CTL1550) is coupled to the silicon nitride chip by using a pair of lensed fibers. The overall insertion loss is usually −3 dB per facet.

When the laser is continuously frequency scanned, free from mode hopping, it features a number of cavity resonances, in terms of the transmitted power trace as the function of the scan time. In the meantime, a fraction of the laser is simultaneously coupled to a referenced fiber-loop cavity that is known to have a free spectral range (FSR) of 20MHz, and a similar power trace showing a number of cavity resonances is recorded as well. Therefore, both power traces can be calibrated and mapped to the frequency axis, in which each resonance of the microresonator is estimated for the resonant frequency and for the resonance linewidth (in the unit of frequency as well). In mathematics, the resonant frequency $\omega_{\mu }$ (as a function of the mode index μ) can be expressed in a Taylor series, and the dispersion profile is extracted as (which is also called the integrated dispersion)
(1)$$D_{\mathrm{int}}\left(\mu \right)=\omega_{\mu }-\omega_{0}-D_{1}\mu =\sum_{m\geq 2}^{}\frac{D_{m}\mu^{m}}{m!}$$
where $\omega_{0}$ indicates the frequency of a central resonant mode (with μ=0) and $D_{m}$ indicates the *m*-th order component of $\omega_{\mu }$ with respect to the central mode. In particular, $D_{1}/2\pi$ corresponds to the FSR of the microresonator, and $D_{2}/2\pi$ indicates the group velocity dispersion (GVD) of the cavity. For anomalous GVD, $D_{2}>0$.

Even though there is certain dispersion in the silica fiber as well as in the fiber loop cavity, the fiber dispersion (D=18ps/(nm·km)) is usually much smaller than that of the silicon nitride waveguides (Typ. D>100ps/(nm·km) at 1550nm). Therefore, the accuracy of the dispersion characterization with a fiber loop cavity would only be weakly affected. As a result, the transmitted power trace of the microresonator (in the transverse ${\mathrm{TE}}_{00}$ mode) and of the fiber-loop cavity is shown in Figure 1c and in Figure 1d, respectively. In the range from 188–200 THz, more than 100 resonant modes of the microresonator are captured and analyzed. Each resonance is Lorentzian-fitted such that its central frequency (i.e., the resonant frequency) and the linewidth can be extracted. The integrated dispersion is then illustrated in Figure 1e, which is compared and shows good agreement with a numerically calculated dispersion profile (COMSOL simulation) based on the same geometric size of the microresonator. This confirms that our dispersion measurement, taking the reference from a fiber-loop cavity, is of decent accuracy.

The estimated FSR of the microresonator is 115 GHz, and the GVD is $D_{2}=1.65\;\mathrm{MHz}$ at a frequency of 193 THz. A profile of the GVD over frequency is also indicated by the COMSOL simulation (Figure 1f). Moreover, the distribution of the estimated resonance linewidth is also analyzed, cf. Figure 1g, which shows a statistic primary value of ca. 80MHz. This corresponds to a loaded quality *Q*-factor of more than $2\times 10^{6}$. Given the anomalous GVD and decent *Q*-factor, soliton microcombs are expected to be generated in this microresonator.

### 2.2. Soliton Microcomb Generation

We next carried out experiments on the soliton microcomb generation in the above mentioned ${\mathrm{Si}}_{3}N_{4}$ microresonator, via the laser tuning scheme [39], cf. Figure 2a as the pumping setup. The tunable laser intrinsically supports either the piezo-electric control or the current tuning as the way to change the frequency. It is driven by an arbitrary function generator and is programmed on the frequency scan speed as well as on the scan range. The laser power is amplified using an erbium-doped fiber amplifier, before being coupled to the ${\mathrm{Si}}_{3}N_{4}$ chip. The transmitted light after the microresonator is characterized both in the optical spectrum and in the low-frequency rf spectrum.

When the laser power is amplified to ca. 1 W and is frequency down-scanned over a cavity resonance at the wavelength of 1551.36nm, the generated power trace in the microresonator shows a triangular profile together with a stair-like pattern at the ending edge, cf. Figure 2b. The latter is known as the *soliton step*, where dissipative temporal solitons are formed and accessed in the resonator. In experiments, the observed soliton step is as wide as 400MHz, much wider than the laser frequency fluctuation (O(1MHz) in free running). As such, when the laser scan is stopped on the soliton step, soliton microcombs, including both the multi-soliton state (i.e., more than one soliton pulses are circulating in the cavity) and the single soliton state, are deterministically accessed (Figure 2e,f) Indeed, the single soliton shows a typical hyperbolic secant ($sech^{2}$) spectral profile. It features a central frequency shift compared with the pump wave, understood as the Raman induced soliton self-frequency shift [64]. In contrast, if the laser is stopped on the slope of the tri-angular trace, an unstable comb state in the regime of the modulation instability (MI state) is observed (Figure 2d). The low frequency rf spectrum of both the soliton comb and the MI state is illustrated in Figure 2c, which is measured with the photo-detection of the comb. Note that in this thread, the comb is spectral filtered to remove the residual pump power, such as to increase the sensitivity of the rf measurement. As a result, while the comb in the MI state shows a wideband noise figure, the soliton comb is of low noise (close to the noise floor).

### 2.3. Fabrication of Home-Developed Erbium-Doped Mono-Core Fiber

In parallel to the study of chip-scale soliton microcombs, we also spent effort in developing high-performance active-doped gain fiber for the power amplification of the microcomb. Regarding the most popular C-band for optical communications, an erbium-doped gain fiber (Er-fiber) is fabricated.

The preformance of the gain fiber is fabricated by atomic layer deposition (ALD, TFS-200, Beneq Inc., Finland) in conjunction with the modified chemical vapor deposition (MCVD) [78,79] , which is distinguished by the standard process merely by MCVD. In detail, a ${\mathrm{SiO}}_{2}$ porous soot layer is first deposited on the inner wall of the silica substrate tube via the MCVD and then semi-vitrified at a high temperature. Second, ${\mathrm{Al}}_{2}O_{3}$ and ${\mathrm{Er}}_{2}O$ films are deposited by means of ALD, in which deionized water is chosen as an oxygen precursor, and $\mathrm{Al}{\left({\mathrm{CH}}_{3}\right)}_{3}$ and $\mathrm{Er}{\left(\mathrm{thd}\right)}_{3}$ are the precursors for aluminum and erbium ions. Remarkably, ALD would enable uniform growth of the active film with control of the thickness possible at the atomic level, which is key to the deposition of high-concentration and well-dispersed erbium ions in the fiber preform. Next, ${\mathrm{SiCl}}_{4}$, ${\mathrm{GeCl}}_{4}$, and $O_{2}$ are simultaneously introduced into the quartz tube such that a layer of ${\mathrm{GeO}}_{2}-{\mathrm{SiO}}_{2}$ is deposited via MCVD, which would be the core of the fiber with increased refractive index. After the doping process, the substrate tube is collapsed at a high temperature of ca. 2000 ${}^{\circ }$C to form the fiber preform. The preform is then drawn to fiber with a drawing tower.

After the fabrication, the fiber cross-section is assessed with a microscope, and the distribution of the refractive index is measured by means of digital holography [80]. The Er-fiber has a single core-cladding structure with step-index profile. The core diameter is 8.09μm. The refractive index difference between the core and the cladding is approximately $9.57\times 10^{-3}$. Moreover, the emission wavelength, peak absorption, and mode field diameter, along with other specifications, are also characterized and are compared with commercial Er-doped fibers (cf. Table 1), which shows a comparative performance. In particular, the doping concentration is reflected as the peak absorption, which in our Er-fiber is greater than the ER16-fiber but slightly lower than the ER30-fiber.

More generally, the developed fiber fabrication process could potentially support a wide range of materials as active doping elements, including the hybrid doping scheme, and can be transplanted to develop active gain fibers at other wavebands [81].

### 2.4. Full-Wave Amplification of Soliton Comb with Home-Developed Erbium-Fiber

We next investigated the power amplification for the soliton microcomb, using the home-developed Er-fiber. The power amplification setup is shown in Figure 3a. The gain fiber is pumped with a 980nm diode laser, in the forward pumping scheme. The soliton microcomb, with a wavelength span more than 200 nm (3-dB bandwidth 60 nm), serves as a full-wave probe signal, which would better reveal the amplification properties (including the gain bandwidth, gain value, etc.) compared with the single-frequency probing laser.

Currently, we are limited by the power of the pumping laser (which is up to 1.2 W), such that the length of the Er-fiber is reduced to meters level, which is believed to be much shorter than commercial erbium-doped fiber amplifiers (EDFAs). In this context, the optimized fiber length is 3 m, in which the amplified spontaneous emission (ASE) could reach the maximal band range with the highest intensity, see Figure 3b. Figure 3c illustrates the spectra of the soliton microcomb, before and after the power amplification. The wavelength range of the net gain is recorded as 1531–1610 nm. The peak gain value is ca. 23 dB at 1560 nm. After the amplification, more than 40 comb modes (spacing 115 GHz) would reach the power of >−10 dBm, which is considered as the onset power level for WDM. The gain performance at different pumping powers is also assessed, see Figure 3d. The optimized pumping power is ca. 900 mW.

Such a gain performance is also compared with a commercial EDFA (Keopsys, CEFA-C-PB-HP), see Figure 4, whose gain value as well as the gain spectrum is tuned via changing the applied current to the pumping laser. As a result, both amplifiers could support a peak gain value over 20 dB, yet the commercial EDFA would have a better performance especially on the short wavelength edge (at around 1540 nm), which is reasonable with a longer gain-fiber length pumped at a higher power (current). Still, our Er-fiber shows a comparative gain bandwidth at the 10-dB level, whose performance could be even better than the commercial one at the low current.

Therefore, we have demonstrated a soliton microcomb seeded and power-amplified multi-frequency laser readily for telecommunications, among other applications. The performance of our Er-fiber is comparative with commercial products, yet further improvements are required to build a powerful amplifier approaching commercial ones. In addition, the most recently reported scheme to the increase in the power efficiency of the soliton microcomb [72] could also help to provide a more powerful seed than the presented state, which may result in a higher output power level after the amplification.

## 3. Conclusions

In conclusion, we have implemented a power-amplified chip-scale soliton microcomb to reach the onset for applications. The soliton microcomb is enabled in a photonic-chip-integrated ${\mathrm{Si}}_{3}N_{4}$ microresonator, which has a wide wavelength span covering the S-C-L waveband of telecommunication. This soliton microcomb fundamentally servers as an ideal full wave probing the signal for the characterization of gain-fiber-based optical amplification, and as a consequence, its power could also be amplified over a certain range. We developed an erbium-doped mono-core fiber that shows comparative specifications relative to commercial ones, and we demonstrated the microcomb power amplification. More than 40 comb modes after amplification could reach the power of >−10 dBm , and few modes >0dBm.

Our ability of combining both chip-scale soliton microcombs and active-doped gain fiber technology would present access to power-sufficient and broadband multi-frequency lasers that are desired in current WDM-based optical communication systems as well as in next-generation photonic data center.

## Figures and Tables

**Figure 1 micromachines-13-02125-f001:**
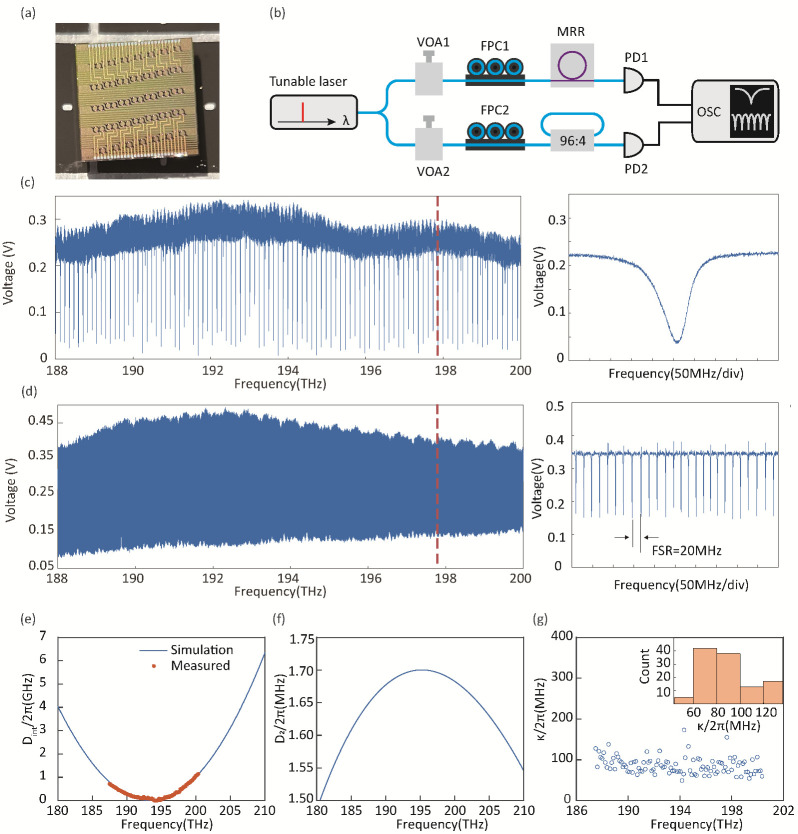
**Characterization of a photonic chip integrated ${\mathrm{Si}}_{3}N_{4}$ microresonator.** (**a**) Picture of a ${\mathrm{Si}}_{3}N_{4}$ microresonator chip. (**b**) Experimental setup for dispersion and linewidth measurement regarding the ${\mathrm{Si}}_{3}N_{4}$ microresonator, where the frequency calibration is via a fiber-loop cavity (FSR∼20 MHz). VOA: variable optical attenuator, PD: photodetector, and OSC: oscilloscope. (**c**,**d**) Transmission trace of both the microresonator and the fiber-loop cavity, upon the frequency tuning of the tunable narrow-linewidth laser. The laser is polarized-controlled to be coupled to the transverse ${\mathrm{TE}}_{00}$ mode in the microresonator, with the power of 1 mW at the input facet of the ${\mathrm{Si}}_{3}N_{4}$ chip. (**e**) Extracted integrated dispersion $D_{\mathrm{int}}/2\pi$ (orange dots) as a function of the resonant frequency, which is compared with COMSOL simulations of the geometric structure of the microresonator. In detail, the ring-waveguide of the resonator has a height of 780nm, a width of 1800nm, and a radius of 220μm. (**f**) The simulated dispersion component $D_{2}/2\pi$ as a function of the resonant frequency. (**g**) The distribution of the resonance linewidths, which are extracted via the Lorentzian fitting of each resonance. The inset is a histogram of the linewidth, which indicates that the statistic linewidth of the microresonator is ca. 80MHz.

**Figure 2 micromachines-13-02125-f002:**
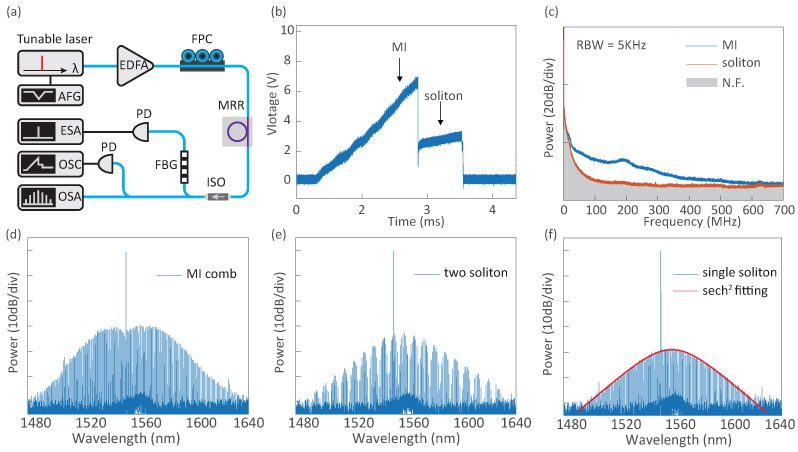
**Soliton microcomb generation in a ${\mathrm{Si}}_{3}N_{4}$ microresonator.** (**a**) Schematic of the experimental setup. AFG: arbitrary function generator, EDFA: erbium-doped fiber amplifier, OSC: oscilloscope, ESA: electrical spectrum analyzer, and OSA: optical spectrum analyzer. (**b**) The generated power trace in the cavity, upon the laser frequency scan over a resonance, and with the power of ca. 1W. The trace shows a triangle profile with a stair-like pattern at the ending edge (also called the soliton step), implying that the soliton microcomb can be accessed. (**c**) Low-frequency RF spectra of both the soliton comb and a comb in the MI state, respectively. Note that the residual pumping wave in the combs is filtered out with a fiber notch filter. (**d**–**f**) Measured comb spectra in the MI state, two-soliton state, and single-soliton state, respectively.

**Figure 3 micromachines-13-02125-f003:**
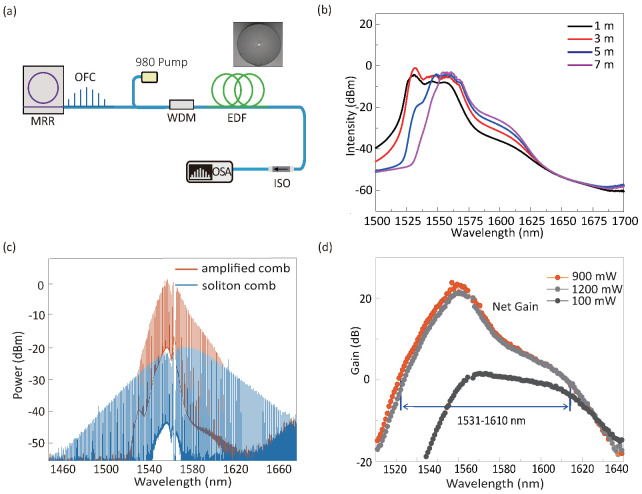
**Soliton microcomb amplification using a home-developed erbium-doped mono-core fiber.** (**a**) Experimental setup for the characterization of the gain fiber, with a soliton microcomb (i.e., the full wave configuration) as the probe signal. The gain fiber is pumped with a 980-nm diode laser. The length of the fiber is ca. 3m. Inset is the cross-sectional view of EDF. (**b**) Comparison of EDF performance in different lengths. (**c**) Power-amplified microcomb spectrum compared with the initial soliton spectrum. The power of the soliton comb at the input is 880μW, and the 980-nm laser power is ca. 900mW. (**d**) Comparison of EDF performance under different pump powers.

**Figure 4 micromachines-13-02125-f004:**
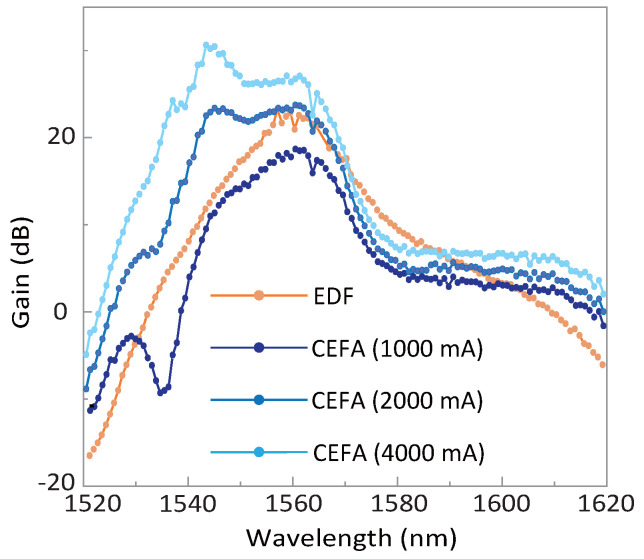
**Gain performance using the commercial EDFA.** The distribution of the net gain is recorded with the applied current at 1000, 2000, and 4000 mA (blue traces). To compare, the gain profile of the home-developed fiber is also presented (orange trace).

**Table 1 micromachines-13-02125-t001:** Specifications of our EDF compared with commercial ones.

Item	ER16-8/125 *	ER30-4/125 *	Our Er-Fiber
Peak core absorption (@ 1530nm)	16±3dB/m	30±3dB/m	22±4dB/m
Mode field diameter (MFD)	9.5±0.8μm	6.5±0.5μm	6.2±0.5μm
Numerical aperture (NA, Nominal)	0.13	0.20	0.16
Cut-off wavelength	1100–1400 nm	890±90nm	1770nm

* Data from Thorlabs, Inc., Newton, NJ, USA.

## Data Availability

Any data related to this work are available from the corresponding authors upon reasonable request.

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
