# Peer review of "Soliton Microcomb on Chip Integrated Si3N4 Microresonators with Power Amplification in Erbium-Doped Optical Mono-Core Fiber"

_micromachines, 2022, doi:10.3390/mi13122125_

Round 1

Reviewer 1 Report

Authors use a commercial SiN4 microring resonator chip to generate Kerr comb with 115 GHz comb spacing. Kerr comb in a single soliton mode has low amplitude and spectral noise, but suffers from low power of comb spectral lines. Authors demonstrate that they could boost comb lines by erbium doped fiber amplifier. Authors have produced erbium doped fiber with high erbium concentration from their in-home-fabricated preform in a fiber drawing tower and achieve best results with a 3 m long gain fiber. As a result more than 40 comb lines are boosted to power level > -10 dBm that is power level necessary for wavelength division multiplexing telecommunications. In the present manuscript authors have not demonstrated the use of their comb in telecommunication data transfer.

Very extensive list of references 78 sources.

1) Referee did not find description of laser incoupling method to the chip. Is it “but-to-butt coupling” or “lensed fiber coupling”, or something else?

2) Line 92 ”fact” à ”facet”

3) About formatting. In figures axis captions many places there is missing space after axis name and units for example ”Frequency(THz)”. Could be changed to ”Frequency (THz)”.

4) Also many references in the text appear without spaces from words, for example, in line 14 telecommunication[1-5] and in many other places throughout the text, see line 95 ” … tunable laser spectroscopy[76,77].

5) Authors say that microring resonator dispersion is calibrated via a fiber loop with 20MHz FSR. Referee could not find in the text if the fiber loop had some dispersion that was  taken into account. May be the authors could add some sentence to clarify. In other groups a femtosecond fiber-based optical frequency comb with strictly equidistant lines is used for calibration, but it is a piece of an expensive equipment not available in many labs, so a fiber loop method sounds very attractive.

5) Line 149 or In Figure 2 caption  ”rf” is not explained that it means radiofrequency and, probably, could be abbreviated with capital letters ”RF”.

6) In Figure 2 (b) and (c) abbreviation is M.I. but in caption MI. Referee would keep MI in all places.

7) Line 192.  ”Servers” à ”serves”.

Author Response

We appreciate reviewers for their comments.

Here we have made a point-to-point reply to all the comments, as shown in the attachment.

Reviewer 2 Report

This paper proposes an erbium-doped gain fiber that can amplify the Soliton micro combs that are produced using Silicon Nitride based Ring resonators. Following are the concerns about this work which otherwise is not bad, I believe that through this the quality of the manuscript can be improved. 

1) Why did you choose this particular platform for soliton generation?

2) A comparison with other platforms would improve the manuscript and enable the reader to evaluate the advantages of the technique better

3) A detailed diagram of the fabrication steps of grain fiber along with its final structure is missing.

4) It is not a review paper, so reducing the number of references, especially where there are duplications would be better

Author Response

(The authors gave the same response as above.)

Reviewer 3 Report

The authors experimentally demonstrated microcomb amplification in a home-made Er-doped fibers. The soliton microcomb is generated in an integrated Si3N4 microresonator chip, which serves as a full-wave probing signal for power amplification. After the amplification, more than 40 comb modes with 115-GHz spacing reach the onset power level of 8 > −10 dBm. Soliton microcomb generation in Si3N4 microrings is a sufficiently mature technology. There are a lot of works on this topic, including by authors of the reviewed manuscript (these papers on SiN-microcombs are cited in the manuscript). The idea of comb amplification for further using is also not novel. Experimental demonstrations of microcomb amplification in Er-doped fibers were previously reported. For example, in [9] the authors generated a soliton comb in a Si3N4 integrated microresonator, amplified this comb in Er amplifier and demonstrated its application for data centers. In the reviewed manuscript the authors used a home-made Er fiber fabricated by using a standard technique. Fiber parameters are not reported; its characterization is also missed. The author showed only an example of amplified spectrum in a 3-m fiber at a fixed pump power without any optimization. This work is not a proof-of-concept, so, research details are necessary. The authors try to claim, that their Er fiber is better than commercial ones, but the following statement of the authors is not justified: “It worth noting that given the high-rate erbium-doped fiber, the required power of the 980 nm laser, as well as the fiber length, is much reduced in our system compared with commercialized EDFA”. It is not clear why the home-made fiber is really better and what are the reasons. I believe that the experimental results are technically correct and reliable, but they have no novelty and originality, so the paper cannot be recommended for publication. 

Author Response

(The authors gave the same response as above.)

Round 2

Reviewer 3 Report

The authors improved the paper and reply satisfactory to my concerns. The novelty has been clarified and research part was extended. Thus, now I can recommend the paper for publication.